# Slow Interstitial Fluid Flow Activates TGF-β Signaling and Drives Fibrotic Responses in Human Tenon Fibroblasts

**DOI:** 10.3390/cells12172205

**Published:** 2023-09-04

**Authors:** Cornelius Jakob Wiedenmann, Charlotte Gottwald, Kosovare Zeqiri, Janne Frömmichen, Emma Bungert, Moritz Gläser, Jeanne Ströble, Robert Lohmüller, Thomas Reinhard, Jan Lübke, Günther Schlunck

**Affiliations:** Eye Center, Medical Center, Faculty of Medicine, University of Freiburg, Killianstr. 5, 79106 Freiburg, Germanyemma.bungert@uniklinik-freiburg.de (E.B.); ornel.lohmueller@uniklinik-freiburg.de (R.L.); jan.luebke@uniklinik-freiburg.de (J.L.); orneliu.schlunck@uniklinik-freiburg.de (G.S.)

**Keywords:** fibrosis, scarring, myofibroblast, glaucoma surgery, ALK5 inhibitor, extracellular matrix, collagen, fibronectin

## Abstract

Background: Fibrosis limits the success of filtering glaucoma surgery. We employed 2D and 3D in vitro models to assess the effects of fluid flow on human tenon fibroblasts (HTF). Methods: HTF were exposed to continuous or pulsatile fluid flow for 48 or 72 h, at rates expected at the transscleral outflow site after filtering surgery. In the 2D model, the F-actin cytoskeleton and fibronectin 1 (FN1) were visualized by confocal immunofluorescence microscopy. In the 3D model, mRNA and whole cell lysates were extracted to analyze the expression of fibrosis-associated genes by qPCR and Western blot. The effects of a small-molecule inhibitor of the TGF-β receptor ALK5 were studied. Results: Slow, continuous fluid flow induced fibrotic responses in the 2D and 3D models. It elicited changes in cell shape, the F-actin cytoskeleton, the deposition of FN1 and activated the intracellular TGF-β signaling pathway to induce expression of fibrosis-related genes, such as *CTGF*, *FN1* and *COL1A1*. ALK5-inhibition reduced this effect. Intermittent fluid flow also induced fibrotic changes, which decreased with increasing pause duration. Conclusions: Slow interstitial fluid flow is sufficient to induce fibrosis, could underlie the intractable nature of fibrosis following filtering glaucoma surgery and might be a target for antifibrotic therapy.

## 1. Introduction

Glaucoma is one of the most common causes of vision loss worldwide. It is characterized by irreversible optic nerve damage due to the death of retinal ganglion cells [1,2,3]. Intraocular pressure (IOP) is the only modifiable risk factor for this condition. Both drug and surgical options can be considered to lower IOP. The reference standard of IOP-lowering filtering surgery is trabeculectomy, in which a capped incision is made in the sclera to create a valve mechanism, allowing aqueous humor to drain from the anterior chamber under the conjunctiva, thereby lowering intraocular pressure. Excessive postoperative fibrosis with associated occlusion of the surgically created outflow pathway is a common postoperative complication despite the use of antimetabolites such as mitomycin C or 5-fluorouracil [4,5,6]. Various compounds have been proposed to control postoperative scarring after filtering glaucoma surgery but have failed to improve clinical practice. Therefore, a better understanding of the underlying pathophysiology and additional options to safely modulate wound healing in this context is desirable.

The conversion of fibroblasts to myofibroblasts is a key step in all fibrotic processes [7]. Myofibroblasts are necessary for adequate wound healing [8], but usually they disappear in the late stages of wound healing [9]. The persistence of myofibroblasts is associated with increased deposition of extracellular matrix (ECM) proteins [10], leading to tissue fibrosis. Myofibroblasts are characterized by increased contractility and the secretion of ECM proteins [11]. The increased contractility is achieved by the integration of alpha smooth muscle actin (α-SMA) into F-actin fibers [12]. Besides growth factors, such as TGF-β, mechanical stimuli can promote myofibroblast transdifferentiation. These may result from changes in the stiffness of the extracellular matrix or from interstitial fluid flow [13,14,15,16,17,18]. In a seminal study using an elaborate 3D in vitro model, Ng et al., provided evidence for a TGF-β-related profibrotic effect of slow interstitial fluid flow in neonatal human dermal fibroblasts [14,15]. As filtering surgery likely alters interstitial fluid flow in the subconjunctival tissue, we hypothesized earlier that interstitial fluid flow might also contribute to fibrosis after filtering surgery [19]. More recently, Gater et al., reported on the effects of various cytokines and fluid movement on porcine tenon fibroblasts. To elicit fluid movement, a rocker platform was used at five rotations per minute to slowly agitate culture dishes of 2D and 3D cultures for 1 h per day [20]. This study suggested that fluid shear stress stimulates cell proliferation, new collagen formation and F-actin expression, leading to a more defined, elongated cell morphology in porcine tenon fibroblasts. However, by design, a see-saw motion rocker model renders poorly defined multidirectional fluid flow and the fluid percolation achieved in the 3D gels appears very limited. Furthermore, trabeculectomy is likely to induce directional fluid flow over longer periods than 1 h per day. To more closely match the conditions of fluid flow to those in the filtering bleb after trabeculectomy, we adapted a 3D perfusion model published earlier [21] to generate a system in which fluid flow can be precisely adjusted and continuously maintained for several days. To mimic interstitial fluid flow in the filtering bleb after filtering glaucoma surgery, flow rates expected to occur postoperatively were tested, as described in more detail in the methods section. The resulting shear forces were several orders of magnitude lower (2 × 10^−2^ to 2 × 10^−4^ dyn/cm^2^) than the shear forces commonly used to study vascular endothelial cells (1 to 100 dyn/cm^2^) [22,23,24,25]. The aim of the current study was to further characterize the effects of continuous or pulsatile fluid flow on HTF. As the results indicated a role for TGF-β-signaling, we explored the impact of a small molecule inhibitor of the TGF-β-receptor ALK5 (ALK5-I.) on flow-induced changes. The inhibitor had shown antifibrotic effects in HTFs after direct TGF-β1-stimulation [26,27].

The cytokine TGF-β is a key mediator of wound healing and is critically involved in postoperative fibrosis [28]. TGF-β signals are transduced by transmembrane type I and type II serine/threonine kinase receptors (also called activin receptor-like kinase 5 or ALK5 and TβRII, respectively). The activated complex of ALK5 and TβRII phosphorylates small mothers against decapentaplegic 2 (SMAD2) and SMAD3, converting them into transcriptional regulators that complex with SMAD4, but may also convey their signals via non-SMAD signaling pathways [29,30]. At the cellular level, TGF-β drives the conversion of fibroblasts to myofibroblasts [7] and is, therefore, a target of antifibrotic strategies [28,31]. TGF-β refers to three cytokines that together form the TGF-β family. In vertebrates, there are three TGF-β isoforms: TGF-β1, TGF-β2 and TGF-β3. TGF-β1 and 2 can induce conversion into myofibroblasts, while TGF-β3 has been shown to exert myofibroblast-suppressing and antiscarring effects [28,32]. However, a previous prospective multicenter clinical trial to inhibit fibrosis following trabeculectomy using a TGF-β2-specific antibody failed to show a significant effect [33]. TGF-β is deposited by cells in an inactive form in the ECM, where it cannot bind to the receptor [34]. The prodomains of TGF-β and TGF-β form a homodimer with a second prodomain-TGF-β complex. Additionally, the complex is bound to another protein called latent transforming growth factor beta-binding protein 1 (LTBP1). This prodomain-TGF-β-LTBP1 complex is bound to ECM proteins via LTBP1, often to fibronectin [35,36,37]. Activation of TGF-β relies on conformational change in the protein complex by mechanical force, leading to TGF-β release to allow for receptor binding on the cellular surface [38]. Prestrained ECM sensitizes embedded latent TGF-β1 for activation in comparison to unorganized and/or relaxed ECM [39]. A recent study on vascular endothelial cells suggested direct mechanosensitivity of the TGF-β-receptor [40]. The TGF-β receptors belong to the serine/threonine kinase family. Activated TGF-β receptors phosphorylate SMAD family proteins intracellularly, which then enter the nucleus and interact with cofactors to influence gene expression [41].

This study establishes that interstitial fluid flow activates TGF-β-signaling to drive fibrotic changes in HTF.

## 2. Materials and Methods

### 2.1. Reagents

Antibodies raised against the following proteins were used: α-SMA (A2547, Sigma-Aldrich, St. Louis, MO, USA), Fibronectin 1 (FN1) (Sigma F 6140, Sigma-Aldrich, St. Louis, MO, USA), latent TGF-β binding protein 1 (LTBP1) (DF10191, Affinity Biosciences, Cincinnati, OH, USA), glyceraldehyde 3-phosphate dehydrogenase (GAPDH) (MAB374, Chemicon, Merck, Darmstadt, Germany), SMAD2 (86F7, Cell-Signaling, Danvers, MA, USA), phosphorylated SMAD2 (pSMAD2) (138D4, Cell-Signaling, Danvers, MA, USA), secondary antibodies goat against mouse immunoglobulin G (IgG) conjugated with Fluorescein Isothiocyanate (FITC) (115-095-146, Jackson Immuno Research, Cambridgeshire, UK) or Tetramethylrhodamine B isothiocyanate (TRITC) (GtxMu-003-J2RHOX, ImmunoReagents, Raleigh, NC, USA) and peroxidase-conjugated goat anti-mouse secondary antibodies (115-035-003, Jackson ImmunoResearch, Cambridgeshire, UK). Furthermore, phalloidin-FITC (P5282, Sigma-Aldrich, St. Louis, MO, USA) was applied. For antibody concentrations refer to Table 1. A total RNA purification kit (RNeasy) and RNase-free DNase were purchased from Qiagen (Hilden, Germany), and reverse transcriptase (SuperScript IV) was purchased from Invitrogen (18090050, Karlsruhe, Germany). TaqMan Fast Advanced Master Mix was purchased from Thermo Fisher (00863702, Waltham, MA, USA). Recombinant TGF-β1 was obtained from Pepro-Tech (Cranbury, NJ, USA) and was used at 5 ng/mL (final concentration) in all experiments. An ATP-competitive inhibitor of TGF-β receptor I kinase (ALK5-I.) (CAS No. 446859-33-2) was purchased from Enzo Life Sciences (ALX-270-445, Farmingdale, NY, USA) and Selleckchem (RepSox (E-616452), Sigma-Aldrich, St. Louis, MO, USA). An anti-TGF-β1 antibody (AF-101) and IgY isotype control (AB-101-C) were purchased from R&D Systems (Minneapolis, MN, USA).

### 2.2. Cell Lines

Small tenon biopsy samples were obtained during squint surgery in otherwise ophthalmologically healthy patients after comprehensive information and written consent. The tenets of the Declaration of Helsinki were followed, and an institutional ethics committee (172/18) approval has been granted. Primary HTF were gained as an expansion culture of the human tenon explants and propagated in low glucose (1 g/L D-glucose) Dulbecco’s modified Eagle medium (DMEM) (11880-63, Life Technologies, Carlsberg, CA, USA) supplemented with 10% heat-inactivated fetal bovine serum (10270106, Life Technologies), 100 U/mL penicillin and 100 µg/mL streptomycin (P4333, Sigma-Aldrich) and 2 mM L-Glutamine (25030-081, Life Technologies). For experiments, FBS was added at different concentrations as specified in the text. For all experiments, cells from passages 3 to 8 were used. All experiments were performed at least three times.

### 2.3. Cell Culture and Interstitial Fluid Flow

Two-dimensional flow experiments were performed on µ-slides I Luer 0.8 mm (ibidi GmbH, Gräfelfing, Germany). Fifteen thousand cells were seeded onto the µ-slides under static conditions in 0.2% FBS-containing cell culture medium. After 24 h (h) incubation and before the start of the flow, cell culture medium was exchanged. In the static controls, medium exchange was performed every 24 h. Flow rates were set as indicated for the respective experiments, resulting in shear stress, as shown in Table 2 [42].

Three-dimensional flow experiments were performed in a collagen gel suspension culture. Each gel contained 30,000 human tenon’s fibroblasts in 500 µL of a collagen solution with a final collagen concentration of 2.18 mg/mL (#5005, Advanced BioMatrix, Carlsbad, CA, USA), which was cast in semipermeable 12-well Millicell Hanging Cell Culture Inserts with pores of 1.0 µm diameter (Merck, Darmstadt, Germany). Prior to stimulation, the gel suspension cultures were incubated in a 12-well plate (150628, Thermo Fisher Scientific) for 72 h in cell culture medium containing 1% FBS.

For the flow experiments, the gel-containing inserts and the flow chamber were assembled under sterile conditions and connected to a syringe pump (KDS 220, kdScientific, Holliston, MA, USA) (Figure 1). Subsequently, gels were incubated for an additional 72 h in static starvation medium (0.2% FBS) serving as a control, or under flow conditions in starvation medium percolating the HTF-populated 3D collagen gel. In some experiments, another static control of starvation medium containing recombinant TGF-β1 (5 ng/mL) was added, as TGF-β1 stimulation of fibroblasts is a reference standard for myofibroblast transdifferentiation. One experiment included another static control using a medium with 10% FBS, which served as a control for enhanced growth factor supply, which might occur in flow conditions. The flow rate was set at 666 µL/h or 180 µL/h thus inducing approximately the same rate of flow as estimated to occur at the site of the outflow pathway, leading to shear stresses of approximately 10^−2^–10^−3^ dyn/cm^2^ [43,44]. All cultures were maintained in a humidified 37 °C, 5% CO_2_-incubator for the duration of the experiment. The proper function of the culture system was checked daily.

### 2.4. Immunofluorescence Confocal Microscopy

To assess the effect of fluid flow in the 2D model, F-actin, α-SMA, FN1 and LTBP1 were visualized by confocal immunofluorescence microscopy. Cells on µ-slides were fixed in 4% paraformaldehyde, permeabilized in 0.1% Triton X-100 (SLBV4122, Sigma-Aldrich), blocked in 5% normal goat serum (NGS) (Dianova, Hamburg, Germany) with 0.1% Triton X-100, and labeled with primary antibodies against α-SMA, FN1 or LTBP1 in 1% NGS and 0.1% Triton X-100 in Dulbecco’s Phosphate Buffered Saline (DPBS) for 16 h at 4 °C. Secondary antibodies conjugated with FITC or TRITC were used in DPBS with 1% NGS and 0.1% Triton X-100. Phalloidin-FITC was used to stain the F-actin cytoskeleton. Cells were viewed with a laser scanning confocal microscope (TCS SP8; Leica Microsystems, Bensheim, Germany) and z-stacks with z-distances of 1 µm between each plane were recorded and are shown as projections. Signal intensities of the proteins of interest were quantified using ImageJ 1.53a (Rasband, W.S., ImageJ, U.S. National Institutes of Health, Bethesda, ML, USA).

To explore the morphologic effects of slow interstitial fluid flow on HTFs in the 3D model, we assessed the F-actin cytoskeleton in HTF-populated collagen gels by immunofluorescence confocal microscopy. Three-dimensional collagen gel cultures were removed from the inserts and fixed in 4% paraformaldehyde for 20 min, washed 5 times for 10 min with DPBS, and subsequently blocked and incubated with primary and secondary antibodies as described above, with the exception that Triton X-100 was used at a concentration of 0.5%. The gel was mounted on a microscopy slide with ProLong Glass Antifade Mountant with NucBlue Stain (P36983, Thermo Fisher) and dried overnight before imaging.

Three-dimensional collagen gel culture z-stacks were scanned with z-distances of 2 µm between each plane. Three-dimensional reconstruction of the cell surfaces was performed with Imaris 9.5 software (Bitplane, Zurich, Switzerland).

### 2.5. Gene Expression Analysis

Collagen gel 3D cultures were removed from the inserts, washed with DPBS and digested with 2 mg/mL collagenase D (11088858001, Roche, Basel, Switzerland) in DPBS at 37 °C on a shaker (900 rotations per minute) for approximately 40 min until the gel structure was entirely dissolved. Collagenase activity was stopped by adding EDTA. Subsequently, the cell suspension was centrifuged for 10 min at 500 g to receive a cell pellet, which was lysed using an RNeasy Mini Kit and QIAshredder homogenizer (Qiagen, Hilden, Germany) according to the manufacturer’s recommendations and treated with DNase I (EN0521, Thermo Fisher Scientific) for 15 min at room temperature during mRNA extraction. The final mRNA concentrations of samples were quantified by NanoDrop-1000 (Thermo Fisher Scientific) and samples stored at −20 °C. First-strand cDNA was synthesized within 24 h after mRNA extraction by SuperScript^®^ IV Reverse-Transcriptase (18090010, Thermo Fisher Scientific) at 52 °C for 10 min.

qPCR was conducted with triplicates in a 96-well microtiter plate (LightCycler^®^ 480 Multiwell Plate 96, white, Roche) capped with ultraclear Optical Flat 8-Cap Strips (TCS 0803, Bio-Rad) with TaqMan Universal PCR Master Mix (4364340, Thermo Fisher Scientific) using the specific commercially available TaqMan sets of primers and probes listed in Table 3 in a LightCycler^®^ 96 (Roche). Cycling conditions were initial denaturation at 95 °C for 25 s, followed by 40 cycles consisting of a 3-s denaturation interval and a 30-s interval for annealing and primer extension at 60 °C. Cq-values for each sample were estimated by LightCycler^®^ 96 Software Version 1.1.0.1320 (Roche). Cq-value averages of sample triplicates were calculated and those of target genes were subtracted from those of reference gene β2-microglobulin (*B2M*), serving as a normalization standard to quantify ΔCq. ΔCq of the target genes in experimental conditions was compared to static controls and relative gene expressions (RGE) calculated. Statistical analysis was conducted using the R software version 4.0.1 [45]. Simultaneous tests for general linear hypotheses were performed by “glht” (multcomp). For analysis of variance, we used “aov” (stats). For multiple comparisons of means, we used Dunnett contrasts. *p*-values below 0.05 were considered statistically significant.

### 2.6. Western Blot

Steps performed to obtain a cell pellet from the 3D collagen gel cell cultures were as described in the gene expression analysis section. The cell pellet was resuspended in T-PER buffer (243205, Sigma-Aldrich) with added protease and phosphatase inhibitors (04906845001 and 04693159001, Roche), shortly vortexed and incubated on ice for 15 min. Subsequently, the lysate was resuspended and centrifuged for 5 min at 12,000 g, and the supernatant containing the protein extracts was boiled after adding ¼ sample volume of 4× Laemmli sample buffer (1610747, Bio-Rad, Feldkirchen, Germany) with 10% β-mercaptoethanol (BCBQ7289V, Sigma-Aldrich) and subjected to SDS-polyacrylamide gel electrophoresis.

Proteins were transferred onto polyvinylidene difluoride membranes. Membranes were activated in methanol, blocked in 3% BSA in TBST (pH 7.4, 0.1% Tween 20) for 30 min, incubated with primary antibody overnight at 4 °C, with a peroxidase-conjugated secondary antibody for 60 min at room temperature. After each incubation step, membranes were washed in TBST four times for 10 min, respectively. Peroxidase was visualized by enhanced chemiluminescence (Pierce ECL Plus Western Blotting Substrate, Thermo Fisher Scientific) and signal intensities were measured using a Fusion FX system (Vilber, Marne-la-Valée, France) at appropriate times.

## 3. Results

### 3.1. Slow Interstitial Fluid Flow Changes HTF Morphology in 3D Cell Culture 

Under flow conditions, HTFs presented an intensified F-actin cytoskeleton, pronounced pseudopodia and slight proliferation compared to the static negative control (Figure 2A). TGF-β1 was applied to static cultures as a profibrotic positive control and induced the greatest increase in F-actin fibers, both in number and intensity, somewhat wider cell bodies and a more branched cell morphology than were observed in static controls. HTFs incubated in a medium containing 10% FBS showed an elongated, thin morphology with more unidirectional and fewer dendritic cell bodies. No significant change in cell count of flow and TGF-β1-stimulation compared to static controls was observed, but stimulated cells showed a larger surface/nucleus and volume/nucleus ratio, arguing for a more elongated and branched shape of the nonproliferative HTFs (Figure 2B).

### 3.2. Fluid Flow Promotes Fibrotic Changes in the 2D Model Depending on Flow Rate

In the 2D model, untreated control HTF showed predominantly few and thin, weakly stained F-actin fibers and an elongated, thin spindle-shaped cell morphology in IF (Figure 3A). Cellular FN1 was diffusely distributed without significant bundle formation. Fibroblasts exposed to fluid flow showed an enhanced F-actin signal, characterized by an increase in the number and thickness of F-actin fibers. HTF morphology appeared more compact and less elongated compared to static controls. Flow also induced a stronger signal for FN1, which appeared more organized and aligned with actin-stress fibers. The flow-induced changes appeared dose-dependent, with a peak at 150 µL/h compared to 58 µL/h and 340 µL/h. F-actin fiber intensity was lower at a flow rate of 340 µL/h compared to 150 µL/h. TGF-β1 stimulation was used as a positive control to assess the capacity of the cells to generate a fibrotic response. It induced pronounced F-actin stress fiber formation with fewer but thicker fibers and a trapezoid morphology. TGF-β1 stimulation had a similar but more distinct effect on FN1 compared to flow. Quantification of the signal intensity of F-actin and FN1 revealed the strongest effect to be induced by a flow rate of 150 µL/h (Figure 3B). Regarding α-SMA, untreated cells expressed little α-SMA, which was predominantly localized in the perinuclear region. TGF-β1 induced α-SMA expression and recruitment to F-actin stress fibers in most cells. Under flow conditions, α-SMA distribution was less focused on the perinuclear region, but no recruitment to actin stress fibers was detected (Appendix A). In the 3D model, a slight increase in α-SMA protein compared to static controls was observed by western blot, but this effect was less pronounced than after stimulation with TGF-β1 (Appendix A). 

### 3.3. Flow Pauses Reduce the Flow Effect in the 2D Model in a Time-Dependent Manner

In the previous experiments, we investigated the influence of continuous interstitial fluid flow on HTFs in a 2D and 3D model. However, after filtering surgery, aqueous outflow might have a pulsatile or even gushing character. To keep the total delivered volume of 10.8 mL over 72 h constant, we increased flow rates with decreasing flow duration. In the 2D model, perfusion was either continuous [150 µL/h], or in pulses of one minute followed by pauses of 1, 3 or 6 h. The flow rates were increased to 9.000, 27.000 or 54.000 µL/h respectively. Slow continuous fluid flow at 150 µL/h for 72 h increased the density of intracellular F-actin stress fibers and the deposition of aligned bundled cellular FN1 in the 2D model (Figure 4A). Pulsatile perfusion of the same total volume over 72 h with perfusion pauses of 1, 3, or 6 h elicited a weaker effect. Here, a decrease in perfusion-specific morphologic changes was evident with increasing pause length. Quantification of the signal intensity of F-actin and FN1 revealed a declining effect with a longer break duration (Figure 4B).

### 3.4. Fluid Flow Leads to Increased Expression of Fibrosis-Associated Genes in the 3D Model

We examined the effect of slow interstitial fluid flow on the expression of selected genes in the 3D model. Expression of collagen type I alpha 1 chain (*COL1A1)* and transforming growth factor beta 1 (*TGFB1)* was significantly increased by flow compared to static controls (Figure 5). Transforming growth factor beta 2 (*TGFB2)* was significantly downregulated. Alpha smooth muscle actin (*ACTA2)*, connective tissue growth factor (*CTGF)*, *FN1* and transforming growth factor beta 3 (*TGFB3)* showed a trend towards increased expression (not statistically significant). In TGF-β1-stimulated HTF, which served as a positive control, *COL1A1*, *CTGF*, *FN1* and *TGFB1* were significantly more expressed. *ACTA2* and *TGFB3* showed a trend towards increased expression (Figure 5). *TGFB2* remained stable. Although the effect of flow on gene expression was less pronounced than the effect of TGF-β1-stimulation, the pattern of gene upregulation was similar. *TGFB2* was significantly downregulated by flow but stable with TGF-β1 stimulation.

### 3.5. Slow Interstitial Fluid Flow Promotes LTBP1 Accumulation in the 2D Model

All TGF-β isoforms are secreted in association with latent TGF-β binding proteins (LTBPs), which anchor the molecule to the ECM and regulate its release and activation. We therefore assessed the distribution of LTBP1 as a possible indication of flow-induced effects on the TGF-β pathway. Under static conditions, mainly perinuclear LTBP1 was observed. Under flow conditions, the accumulation of LTPB1 increased (Figure 6).

### 3.6. Flow-Induced Effects on HTF Are Reduced by a Small Molecule Inhibitor of TGF-β RI Kinase

In the 2D model, the addition of an ATP-competitive inhibitor of TGF-β RI kinase (ALK5-I.) showed a slight baseline decrease in F-actin fibers and FN1 in static controls and the effects of flow on the F-actin cytoskeleton and FN1 bundle formation were significantly reduced (Figure 7A,B). In the 3D model, expression of actin-beta (*ACTB)*, *COL1A1, FN1* and *TGFB1* was significantly increased by flow compared to static controls and *ACTA2, CTGF, Serpin Family E Member 1* (*SERPINE1)* and *SMAD7* showed a trend towards increased expression (not statistically significant) (Figure 7C and Appendix A). Expression of these genes did not change significantly by the addition of ALK5-I to static controls. However, the flow-induced increase in expression of *COL1A1, FN1* and *TGFB1* was abolished by ALK5-I. and *TGFB3*-expression fell below baseline control levels. The similarity of expression pattern changes induced by TGF-β1 and fluid flow (Figure 5) as well as the effects of an ALK5-inhibitor (Figure 7A–C) prompted us to study the phosphorylation of SMAD2, the downstream effector of TGF-β1 signaling. Western blot analysis revealed a clear flow-induced increase in pSMAD2 and a prominent loss of SMAD2 phosphorylation in the presence of an ALK5-inhibitor (Figure 7D).

A previous study on dermal fibroblasts [14] had reported inhibition of flow-induced effects by a TGF-β1-binding antibody. Along these lines, we assessed the effect of a TGF-β1 binding antibody (4 µg/mL) on HTF in 2D flow, which also rendered reduced signals for F-actin and FN1. However, the same was observed using an IgY isotype control antibody (Appendix A), an observation that was not reported in [14].

## 4. Discussion

Fibrosis poses the greatest challenge to the long-term therapeutic success of filtering glaucoma surgery. There is no escaping the fact that filtering bleb scarring is a complex process with several likely contributing factors, including sutures, immune cells, cytokines and growth factors derived from the blood, surrounding tissue and aqueous humor [19]. Several studies addressed the direct effect of TGF-β on HTF. In this study, we show that slow fluid flow is sufficient to induce fibrotic changes in HTF by activating TGF-β-signaling. These effects are sensitive to flow rates and flow patterns and are blocked by a small-molecule inhibitor of TGF-β-receptor I kinase.

Interstitial fluid flow exerts shear forces that are several orders of magnitude smaller than intravascular fluid flow. In vascular endothelial cells, shear stress sensing has been shown to involve several cellular structures and signaling pathways, such as the glycocalyx, integrin signaling, cell-cell adhesions and the actin cytoskeleton [46,47]. In addition, ion channels have a role in the detection of small shear forces, as has been reported in mesenchymal cells of the ocular outflow tract [48,49]. Mechanotransduction by osmotic pressure or tissue compaction involves similar components; however, their relative contributions are cell-type-, tissue- and stimulus-dependent.

Ng et al., had observed an alignment of dermal fibroblasts perpendicular to the direction of fluid flow [14], whereas we detected no significant alignment of tenon fibroblasts in our model. This might be attributed to differences in the experimental 3D flow setting, as exact cell orientation is difficult to measure in our model due to the lack of a distinct spatial reference point in the mounted specimens. However, we did not detect a specific cellular orientation in the 2D model either. Another factor could be the differences in flow rates and resulting shear stress, which were significantly lower in our experiments designed to mimic aqueous outflow.

Myofibroblasts are characterized by increased contractility, expression of α-SMA and secretion of ECM proteins, such as fibronectin and collagens. The increased contractility is associated with the integration of α-SMA into F-actin fibers. In dermal fibroblasts, flow led to increased α-SMA-expression [14]. This is in line with our data on HTF, which indicate a flow-induced increase in α-SMA expression as detected by immunofluorescence and western blot analyses. In contrast to TGF-β1-stimulation, slow fluid flow did not promote the incorporation of α-SMA into large F-actin stress fibers, although the F-actin fibers were clearly more enhanced by fluid flow as compared to static controls. This may indicate that direct stimulation by TGF-β commonly used to study myofibroblast transdifferentiation in vitro, elicits different or harsher responses than the shear stress exerted by fluid flow in our system.

We also detected a flow-induced increase in fibronectin protein deposition (Figure 4) and enhanced transcription of *ACTA2*, *CTGF*, *FN1*, *COL1A1* and *TGFB1* in a similar pattern but on a lower level than elicited by TGF-β1 stimulation (Figure 5). In contrast, transcription of *TGFB2* decreased as compared to controls. Furthermore, immunofluorescent staining of LTBP1 in the 2D model indicated enhanced expression of this protein by HTF in response to slow fluid flow. These data are compatible with flow-induced deposition of TGF-β as had been suggested by Ng et al., based on immunofluorescence staining for TGF-β and inhibitory effects of a TGF-β-binding antibody [14]. We also tested a TGF-β1-binding antibody in our 2D flow model to block autocrine TGF-β1-stimulation and observed a reduction in flow-induced changes; however, this effect was also found in IgY isotype controls, arguing against a specific effect of the TGF-β-binding antibody (Appendix A). In this study, we focused on inhibiting TGF-β1 with antibody binding, as this could be a potential strategy to prevent flow-induced fibrosis in patients. However, to rule out the possibility of autocrine TGF-β signaling, genetic silencing or knockout experiments would be necessary. Genetic targeting of TGF-β in tenon fibroblasts seems a possibility but may be insufficient if other sources of TGF-β are relevant. In addition, care must be taken not to interfere with the tumorsuppressive effects of TGF-β in epithelial cells, which would require a very specific targeting approach. Furthermore, in a study on vascular endothelial cells, a TGF-β-independent activation of the ALK-5 receptor by flow has been suggested [40]. Therefore, a transient modulation of TGF-β receptor activation using small molecule inhibitors appears to be an attractive approach.

To further elucidate a possible role of TGF-β signaling in flow-induced fibrosis, we used RepSox, a small molecule inhibitor of the type I TGF-β receptor ALK5, in our 2D and 3D models. Flow-induced FN1 deposition and changes to the actin cytoskeleton were reduced, and transcription of fibrosis-related genes (*ACTA2*, *FN1*, *COL1A1*, *CTGF* and *TGFB1*) was attenuated by RepSox. Furthermore, Western blot revealed that slow fluid flow enhanced Smad2 phosphorylation in the 3D model. These data strongly support the crucial role of the TGF-β signaling pathway in HTF flow responses. The source of TGF-β in this setting is unclear. Our qPCR data revealed a flow-induced increase in *TGFB1* and *TGFB3* transcription, suggesting an auto- or paracrine mechanism. Autocrine stimulation and a possible presence of TGF-β in the cell culture medium could contribute to the finding that an ALK5-inhibitor also reduced the baseline expression of fibrosis-related genes. Unspecific effects of small molecule inhibitors are a frequent concern. For RepSox, the half maximal inhibitory concentration (IC50) is 4 nM for ALK5 autophosphorylation, as compared to an IC50 of more than 16 µM for nine other kinases [50]. Since we used a final concentration of 100 nM, off-target effects should play a minor role in our study.

The valve mechanism created by trabeculectomy is likely to result in pulsatile rather than continuous outflow of aqueous into the filtering bleb. We observed a decreasing flow effect an with increasing pause duration between pulses. Transferring these results to the clinic, a controlled bulbar massage after filtering surgery might be useful, at least with regard to scar formation.

In conclusion, our in vitro data raise the possibility that the desired effect of filtering surgery, namely, the percolation of aqueous humor into the subconjunctival space, may itself promote fibrosis. A better understanding of prevailing flow rates, flow patterns and their respective effects could help improve clinical management strategies. One might speculate that fluid flow-induced auto- or paracrine expression of TGF-β1 may have contributed to the insufficient efficacy of a TGF-β2-specific antibody in a well-designed earlier clinical trial [33]. Apart from its profibrotic effects, TGF-β exerts important antiproliferative and immunomodulatory functions, which call for caution with global inhibition of TGF-β signaling. However, ALK5-inhibitors such as Galunsertib have been used systemically in phase 2 clinical trials, e.g., in myelodysplastic disease, with limited side effects [51]. Based on our findings, subconjunctival application of small-molecule ALK5 inhibitors in slow-release formulations may merit further exploration in the context of filtering glaucoma surgery.

## Figures and Tables

**Figure 1 cells-12-02205-f001:**
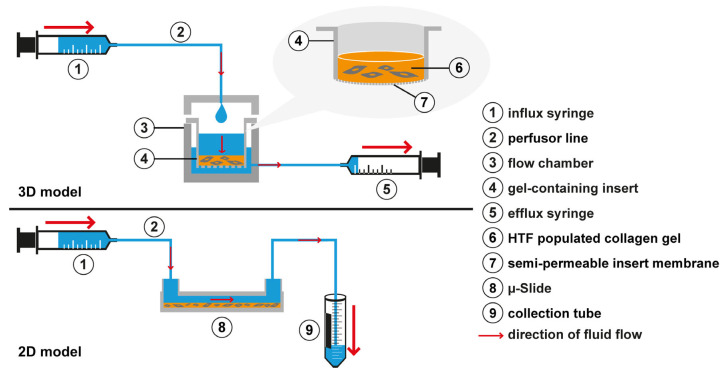
Experimental setup. In the custom-made 3D flow chamber (3), an inlet in the chamber lid was connected to the influx syringe (1) that was mounted on a syringe pump (not shown). Flow is generated following a hydrostatic pressure gradient from the insert (4) through the cell-populated collagen gel (6) and the semipermeable insert membrane bottom (7) into the outer chamber (3). To maintain a hydrostatic gradient, the outer chamber (3) was drained close to the bottom by an efflux syringe (5) aspirating at the same flow rate as the influx syringe. To induce flow in the 2D µ-slide, the µ-slide inlet port was connected by a perfusor line (2) to an influx syringe (1). The outlet of the µ-slide (8) was connected to a second perfusor line that led into a collection tube (9) for the medium.

**Figure 2 cells-12-02205-f002:**
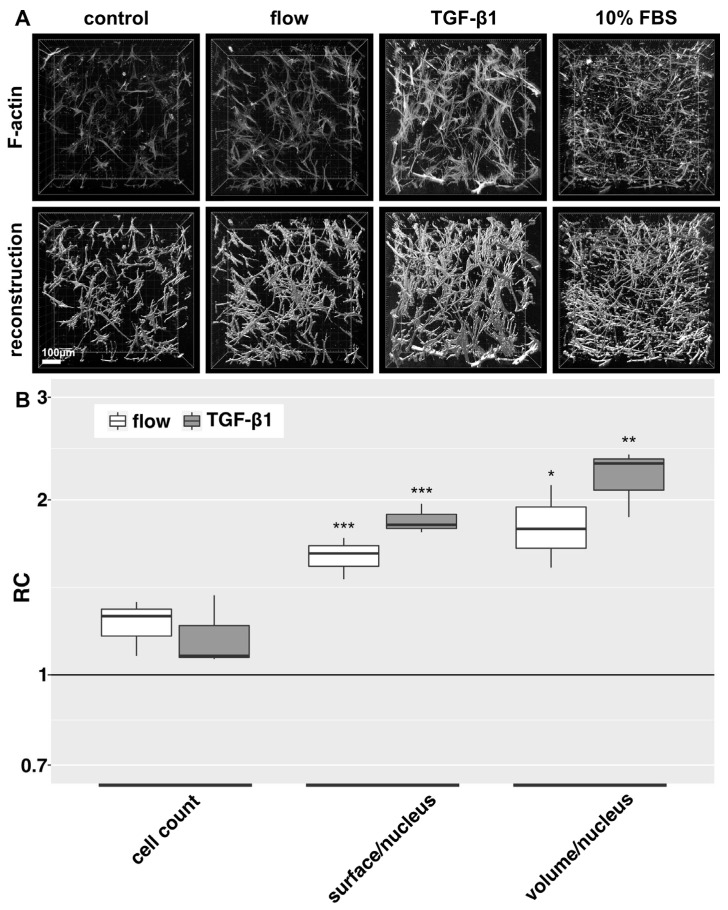
F-actin cytoskeleton and 3D cell reconstructions of 3D HTF cultures exposed to slow fluid flow or control conditions for 72 h. HTF-populated collagen gels were exposed to either 0.2% FBS static medium (control), 666 µL/h flow (0.2% FBS), 10% FBS static medium or 5 ng/mL TGF-β1 (static, 0.2% FBS) for 72 h respectively. (**A**): Confocal image z-stack projections of F-actin stains are presented in the upper row. Three-dimensional reconstructions (lower row) of cell shapes were generated from z-stacks of F-actin stains using Imaris software. Image acquisition and representation are identical among different conditions. Three-dimensional reconstruction of 10% FBS was only performed for one experiment. (**B**). Relative change (RC) of cell count, surface/nucleus and volume/nucleus is displayed for flow and TGF-β1 stimulation. These data were calculated from the confocal image z-stack projections of Figure 2A. Asterisks indicate levels of significance in Dunnett’s *t*-test (* *p* < 0.05, ** *p* < 0.01, *** *p* < 0.001).

**Figure 3 cells-12-02205-f003:**
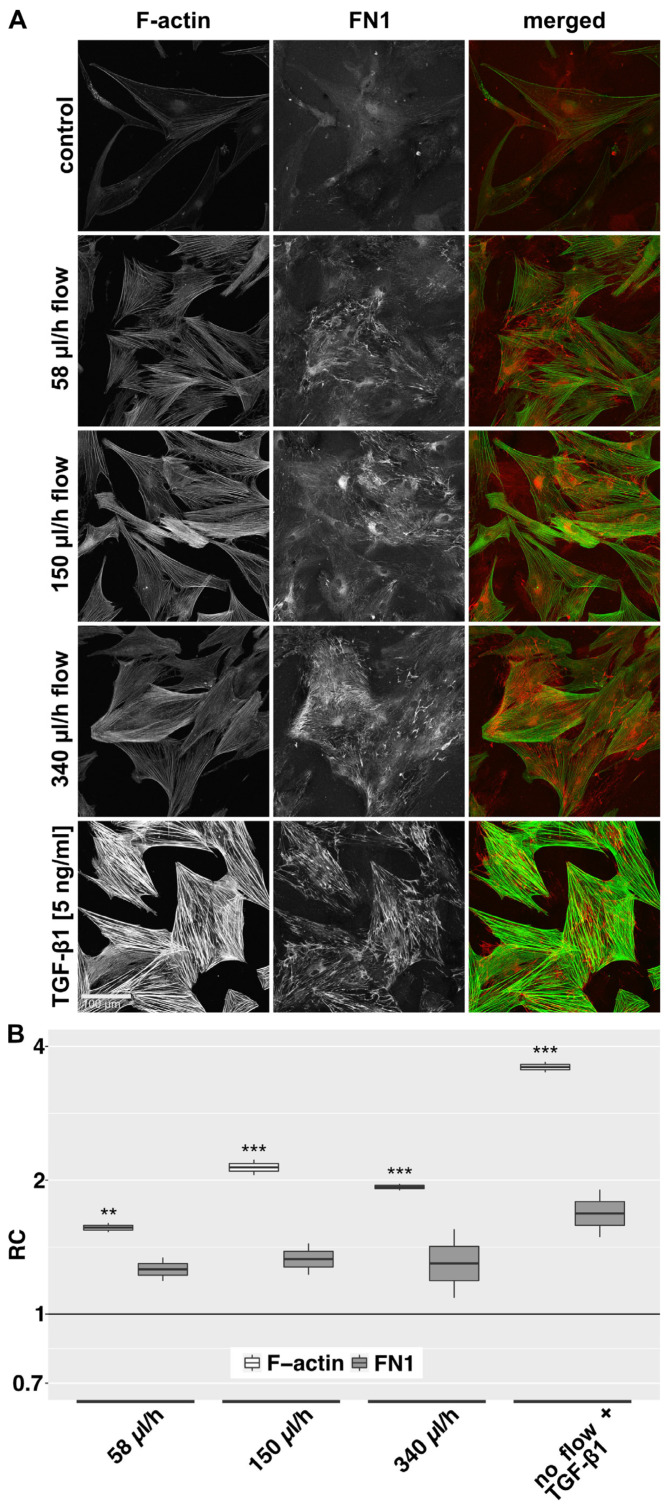
F-actin and FN1 at different flow rates in the 2D model. (**A**): Confocal image z-stack projections of F-actin and FN1 in HTFs exposed to different flow rates or TGF-β1 (5 ng/mL) are displayed. Cells were preincubated in µ-slides for 24 h in starvation medium (0.2% FBS), then perfused for 72 h with the respective flow rate or stimulated with TGF-β1 (5 ng/mL) in starvation medium (0.2% FBS). Image acquisition and representation settings are identical for all conditions. The figure is representative of three independent experiments. (**B**): The relative change (RC) in mean signal intensities for F-actin and FN1 in the projected confocal image stacks of three independent experiments, as displayed in Figure 3A, is shown. Image acquisition and representation are identical under different conditions. Asterisks indicate levels of significance in Dunnett’s *t*-test (* *p* < 0.05, ** *p* < 0.01, *** *p* < 0.001).

**Figure 4 cells-12-02205-f004:**
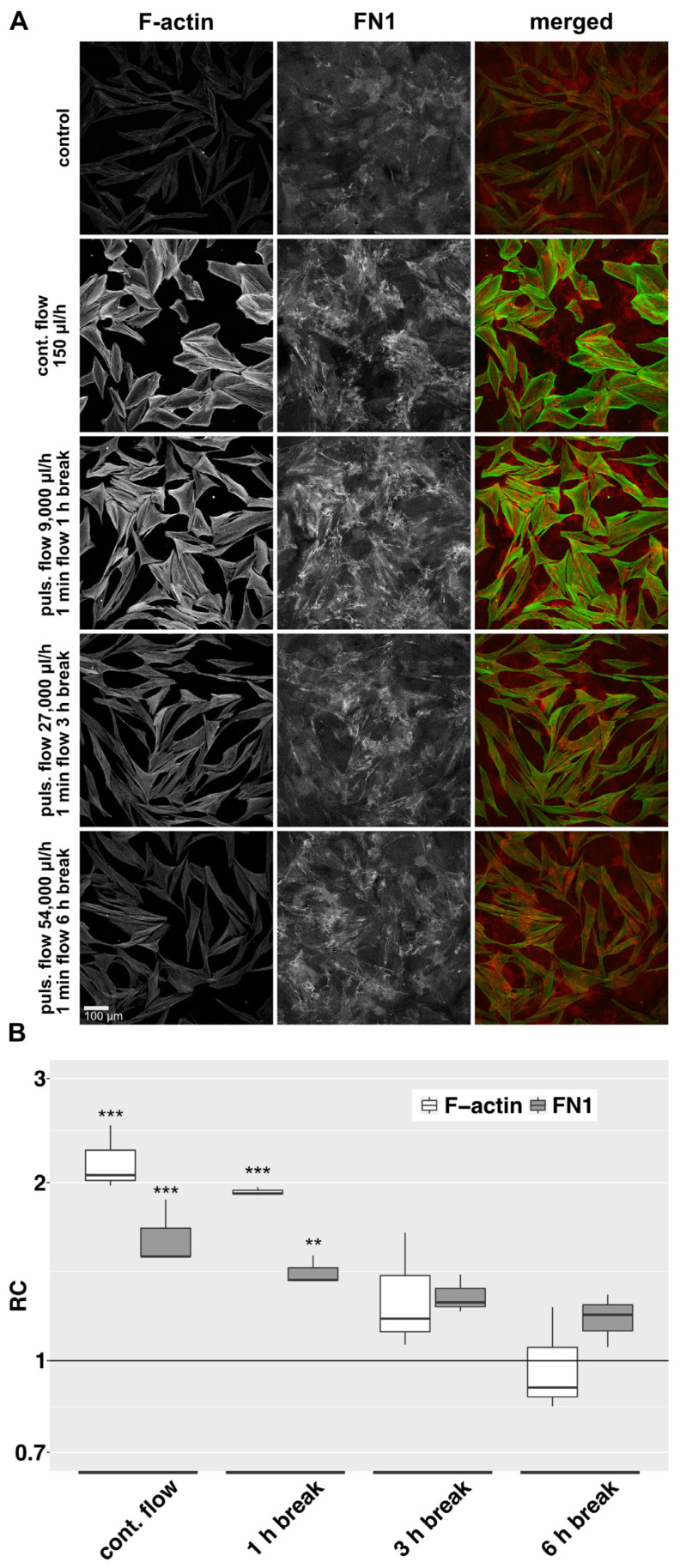
F-actin and FN1 at different pulsatile flow conditions in the 2D model. (**A**): Effects of different pulsatile flows on F-actin and FN1 after 72 h were compared to continuous flow and static conditions in the 2D model. Image acquisition and representation settings are identical for all conditions. The figure is representative of three independent experiments. (**B**): The relative change in the mean signal intensity of F-actin and FN1 in the confocal images of the three independent experiments as displayed in Figure 4A is shown. Image acquisition and representation are identical under different conditions. Asterisks indicate levels of significance in Dunnett’s *t*-test (* *p* < 0.05, ** *p* < 0.01, *** *p* < 0.001).

**Figure 5 cells-12-02205-f005:**
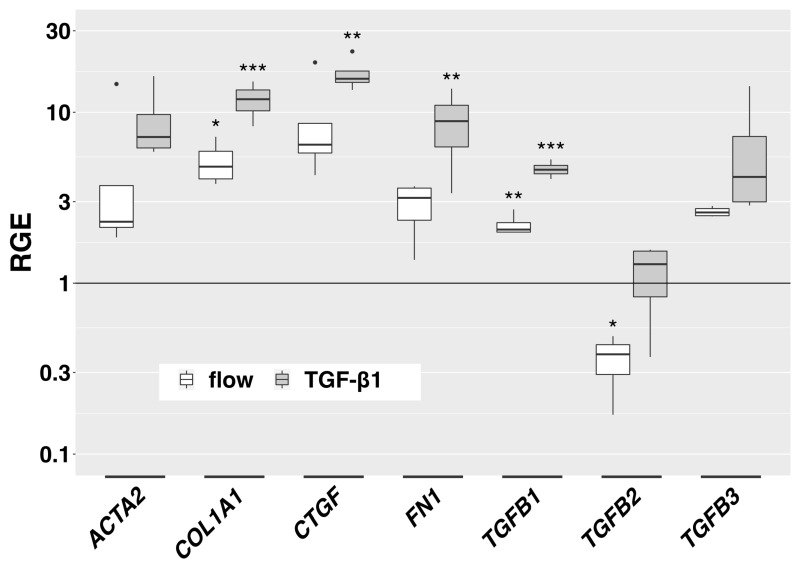
Flow-induced gene expression changes in the 3D model. Relative gene expression of fibrosis-associated genes in the 3D model compared to static controls is displayed for flow conditions (666 µL/h) and TGF-β1-stimulation (5 ng/mL) (72 h respectively, *n* = 4). Asterisks indicate levels of significance in Dunnett’s *t*-test (* *p* < 0.05, ** *p* < 0.01, *** *p* < 0.001).

**Figure 6 cells-12-02205-f006:**
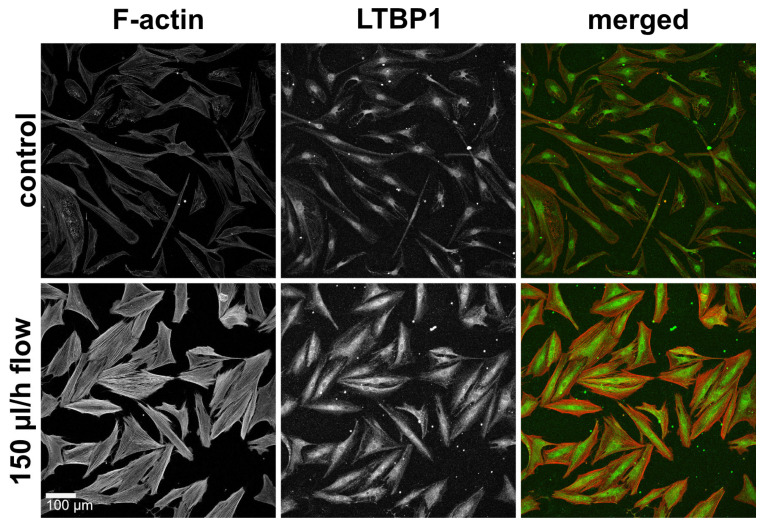
F-actin and LTBP1 under static and flow conditions in the 2D model. Effect of continuous flow (150 µL/h, 72 h) on LTBP1 in the 2D model. Image acquisition and representation settings are identical for all conditions. The figure is representative of three independent experiments.

**Figure 7 cells-12-02205-f007:**
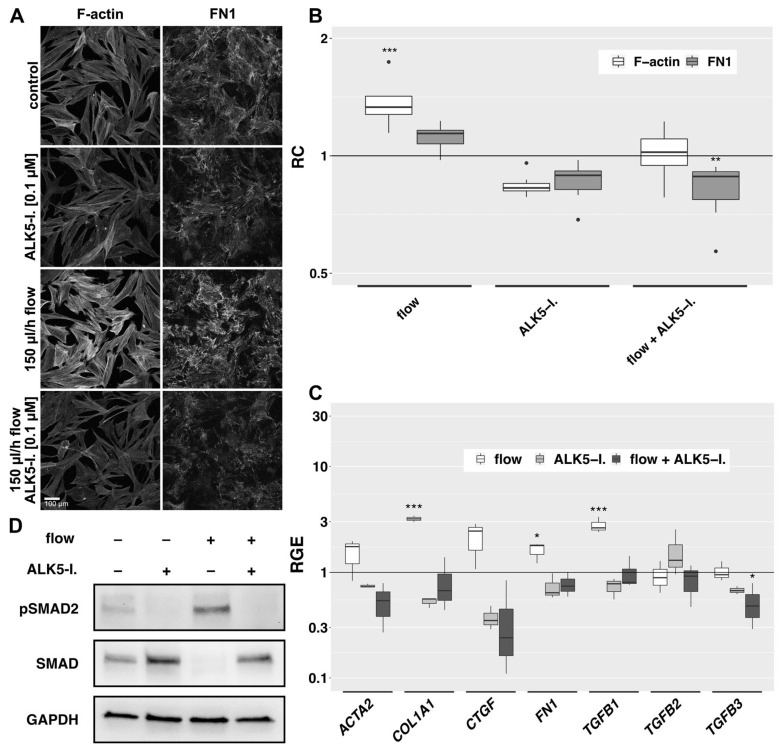
Effects of ALK5-inhibition on the flow-induced changes. (**A**): Effect of ALK5-I. (0.1 µM) on flow-exposed HTF in the 2D model. Cells were preincubated in µ-slides for 24 h in 0.2% FBS medium, then for additional two days incubated either in static or flow conditions in the absence or presence of an ALK5-I. F-actin and FN1 were detected by IF. Image acquisition and representation settings are identical for all conditions. The result is representative of seven independent experiments. (**B**): The relative change in the mean signal intensity of F-actin and FN1 in the confocal images of the seven independent experiments as displayed in Figure 7A is shown, with a region of interest automatically defined by ImageJ. Image acquisition and representation settings are identical for all conditions. Asterisks indicate levels of significance in Dunnett’s *t*-test (* *p* < 0.05, ** *p* < 0.01, *** *p* < 0.001). (**C**): Relative gene expression of fibrosis-associated genes in the 3D model under flow (666 µL/h), static conditions with ALK5-I. and flow conditions with ALK5-I. (0.1 µM, respectively) compared to static controls (72 h, *n* = 4). Asterisks indicate levels of significance according to Dunnett’s *t*-test (* *p* < 0.05, ** *p* < 0.01, *** *p* < 0.001). (**D**): Western blot analysis of protein levels in whole cell lysates of the 3D model under static conditions or flow (180 µL/h) in the absence or presence of an ALK5-inhibitor (0.1 µM, 72 h, *n* = 3).

**Table 1 cells-12-02205-t001:** Antibody concentrations.

Antibody/Phalloidin	Western-Blot Concentration	Immunofluorescence 3D Concentration	Immunofluorescence 2D Concentration
α-SMA	1:1000		1:250
GAPDH	1:5000		
Secondary antibody	1:10,000	1:250	1:250
FN1	1:1000		1:400
Phalloidin-FITC/-TRITC		1:500	1:500
LTBP1			1:200
SMAD	1:1000		
pSMAD2	1:1000		

**Table 2 cells-12-02205-t002:** Flow rates and resulting shear stresses in 2D µ-slides.

Flow Rate	Shear Stress
58 µL/h	0.0002415 dyn/cm^2^
150 µL/h	0.0006246 dyn/cm^2^
340 µL/h	0.0014157 dyn/cm^2^
9000 µL/h	0.0374744 dyn/cm^2^
27,000 µL/h	0.1124232 dyn/cm^2^
54,000 µL/h	0.3372697 dyn/cm^2^

**Table 3 cells-12-02205-t003:** TaqMan primers and probe sets used for qPCR.

Primer	Gene	Company
Hs00426835_g1	*ACTA2*	Thermo-Fisher, Waltham, MA, USA
Hs00187842_m1	*B2M*	
Hs00164004_m1	*COL1A1*	
Hs00170014_m1	*CTGF*	
Hs01549976_m1	*FN1*	
Hs00998133_m1	*TGFB1*	
Hs00234244_m1	*TGFB2*	
Hs01086000_m1	*TGFB3*	
Hs00167155_m1	*SERPINE1*	
Hs00998193_m1	*SMAD7*	
Hs01060665_g1,	*ACTB*	

## Data Availability

The data presented in this study are available in the results section. Further data are available upon request from the corresponding author.

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
