# Peer review of "Slow Interstitial Fluid Flow Activates TGF-β Signaling and Drives Fibrotic Responses in Human Tenon Fibroblasts"

_cells, 2023, doi:10.3390/cells12172205_

Round 1

Reviewer 1 Report

This is an interesting study describing the role of slow interstitial fluid in fibrosis. Using an in vitro setup and human tenon fibroblasts, the authors show that fluid has an effect on the morphology and expression of fibrosis-related genes in HTN, and that these changes are reverted by use of ALK-5, an ATP competitive inhibitor of TGFbetaR-I.

The paper is worthwhile, containing some interesting data. However, several formal issues preclude publication in its present form.

MAJOR ISSUES

1)      Fig. 1 needs to be quantified. Also, the effect of flow and TGF-beta on cell proliferation needs to be measured. Looking at the figures, it is difficult to ascertain whether the effects are due to morphological changes or increased proliferation.

2)      Figs. 2 and 3 need to be quantified as well. Does flow have an effect on the expression of beta-actin as well? Also, these figure legends need to mention explicitly that image acquisition and representation are identical among conditions. This is implied, but proper data representation needs this stated explicitly.

3)      The effect of ALK-5 on F-actin assembly and FN1 deposition need to be quantified. Figures indicate the authors are likely correct, but they need to provide properly quantified data.

4)      The authors need to discuss whether genetic targeting of TGF-beta1, which is upregulated by flow and TGF-beta in an autocrine loop, would be enough to negate the effect of slow flow.

5)      The authors need to discuss the effect of slow interstitial flow compared to other types of mechanical stimulation, for example shear stress on monolayers (similar to what endothelial cells are subjected to in venules), osmotic pressure or mechanical compaction of the tissue (as in tumors).

Reviewer 2 Report

The Article entitled 'Slow interstitial fluid flow activates TGF-β signaling and drives fibrotic responses in human tenon fibroblasts by Wiedenmann et al describes the role of TGFbeta signalling in driving fibrotic responses in Human tenon fibroblasts.

The article is very intriguing, well written, and embodies interesting data to support the induction of fibrosis by TGFbeta signalling. However, The authors could improve the article.

1.      Abbreviations are required for the genes described in Figure 5, ACTA2, COL1A1, CTGF etc.

2.      Which antibody for pSMAD was used in Fig 7C ? Please mention it.

3.       Total SMAD blot is required in Fig 7C along with pSMAD.

4.      The authors need to show qPCR data for TGFbeta responsive genes such as SMAD7.

5.      The authors nicely show the activation of fibronectin, can the authors show the expression of epithelial genes such as E-cadherin?

6.      The authors can comment about interplay of EMT in the context of Fibrosis. They could investigate the potential genes regulated by this pathway.

7.      In section 3, The authors need to include about TGFbeta regulated Smad signalling pathways. Below are some relevant articles.

Signaling Receptors for TGF-β Family Members

Carl-Henrik Heldin, Aristidis Moustakas

Non-Smad signaling pathways, Y Mu, SK Gudey, M Landström

Cell and tissue research 347 (1), 11-20

8.      The authors performed the experiments with 10% FBS which is enriched with growth factors, can the authors perform similar experiments with 1% or 0.5% FBS in Media and culture cells. Which could give more insights.

9.      It would be interesting to perform invasion assays with collagen matrix to see if TGFbeta is driving the phenotype observed by the authors.

Round 2

Reviewer 1 Report

Congratulations on a nice study!